# Effects of a Catechol-Functionalized Hyaluronic Acid Patch Combined with Human Adipose-Derived Stem Cells in Diabetic Wound Healing

**DOI:** 10.3390/ijms22052632

**Published:** 2021-03-05

**Authors:** Chang Sik Pak, Chan Yeong Heo, Jisoo Shin, Soo Young Moon, Seung-Woo Cho, Hyo Jin Kang

**Affiliations:** 1Department of Plastic and Reconstructive Surgery, Seoul National University Bundang Hospital, Seoul National University College of Medicine, Seongnam 13620, Korea; iloveps@naver.com (C.S.P.); lionheo@gmail.com (C.Y.H.); moonsy9970@gmail.com (S.Y.M.); 2Department of Plastic and Reconstructive Surgery, Asan Medical Center, University of Ulsan College of Medicine, Seoul 05505, Korea; 3Department of Biotechnology, Yonsei University, Seoul 03722, Korea; ji_soo_@naver.com; 4Biomedical Research Center, Korea University Ansan Hospital, Ansan 15355, Korea

**Keywords:** diabetic wound, hyaluronic acid, biomaterial, adipose-derived stem cells, angiogenesis

## Abstract

Introduction: Chronic inflammation and impaired neovascularization play critical roles in delayed wound healing in diabetic patients. To overcome the limitations of current diabetic wound (DBW) management interventions, we investigated the effects of a catechol-functionalized hyaluronic acid (HA-CA) patch combined with adipose-derived mesenchymal stem cells (ADSCs) in DBW mouse models. Methods: Diabetes in mice (C57BL/6, male) was induced by streptozotocin (50 mg/kg, >250 mg/dL). Mice were divided into four groups: control (DBW) group, ADSCs group, HA-CA group, and HA-CA + ADSCs group (*n* = 10 per group). Fluorescently labeled ADSCs (5 × 10^5^ cells/100 µL) were transplanted into healthy tissues at the wound boundary or deposited at the HA-CA patch at the wound site. The wound area was visually examined. Collagen content, granulation tissue thickness and vascularity, cell apoptosis, and re-epithelialization were assessed. Angiogenesis was evaluated by immunohistochemistry, quantitative real-time polymerase chain reaction, and Western blot. Results: DBW size was significantly smaller in the HA-CA + ADSCs group (8% ± 2%) compared with the control (16% ± 5%, *p* < 0.01) and ADSCs (24% ± 17%, *p* < 0.05) groups. In mice treated with HA-CA + ADSCs, the epidermis was regenerated, and skin thickness was restored. CD31 and von Willebrand factor-positive vessels were detected in mice treated with HA-CA + ADSCs. The mRNA and protein levels of *VEGF*, *IGF-1*, *FGF-2*, *ANG-1*, *PIK*, and *AKT* in the HA-CA + ADSCs group were the highest among all groups, although the *Spred1* and *ERK* expression levels remained unchanged. Conclusions: The combination of HA-CA and ADSCs provided synergistic wound healing effects by maximizing paracrine signaling and angiogenesis via the PI3K/AKT pathway. Therefore, ADSC-loaded HA-CA might represent a novel strategy for the treatment of DBW.

## 1. Introduction

Diabetic wound (DBW) is a broad term describing various pathological conditions that manifest as wounds or ulcers associated with diabetes. Diabetic foot ulcers and other DBWs are common complications in diabetes, occurring in approximately 20% of diabetic patients [1]. DBWs have been associated with hyperglycemia, which often results in diabetic peripheral neuropathy and blockage of peripheral blood vessels, ultimately leading to diabetic foot ulcers [2]. Patients with DBWs often progress rapidly, making treatment challenging and potentially resulting in lower extremity amputation. In addition to diabetic peripheral neuropathy and peripheral vascular disease, several other risk factors of DBW have been identified, including limited joint mobility and foot deformities [2,3]. Chronic inflammation is an important barrier in the treatment of DBW. Inflammation plays a crucial role in wound healing; in DBW, inflammatory responses are delayed, and the wound does not heal. In addition, the balance between collagen production and degradation is disrupted, further impairing the wound healing process [4]. The clinical management of DBW typically involves wound dressing and debridement of necrotic tissues. Numerous recent studies have indicated that modulating extracellular matrix (ECM) synthesis, growth factor release, and vascularization-targeting approaches might be useful in the treatment of DBW.

Many clinical trials with mesenchymal stem cells (MSCs) are currently in progress as they are readily accessible and safer than other types of stem cells. MSCs represent a stable source for experiments or clinical treatment because they can be obtained from various tissues [5]. MSCs also exhibit multilineage differentiation potential and exert various immunomodulatory and paracrine effects [6]. Adipose-derived stem cells (ADSCs), which can be isolated easily and in large numbers from adipose tissue through liposuction surgery and have been tested for various clinical applications, exhibit long-term growth in vitro and can differentiate into various cell types upon induction [7]. Moreover, ADSCs are known to exert paracrine effects that promote tissue regeneration [6].

Hyaluronic acid (HA) is a polysaccharide found in various tissues of the human body, including the joints, cartilage, eyes, and skin [8,9]. HA plays a key role in wound healing by regulating cell proliferation, migration, and differentiation, as well as ECM organization and metabolism. High molecular weight HA inhibits the proliferation and migration of most cell types, while low molecular weight forms of HA (<300 kDa) promote cell proliferation and display angiogenic properties [10,11,12]. HA in the molecular weight range of 150–250 kDa has been shown to have beneficial effects on wound healing by enhancing cell–HA interactions through cell-surface receptors for HA, which activates signal transduction pathways essential for cellular migration and proliferation [13,14]. Therefore, HA has been proposed as a biomaterial for DBW treatment [8]. HA hydrogels are widely used in numerous biomedical and pharmaceutical applications due to their inherent biocompatibility, matrix structure similarity, and drug delivery capabilities [15]. However, the application of injectable hydrogels to treat human diseases remains limited. Furthermore, the incorporation of stem cells or growth factors into hydrogels remains technically challenging. Catechol-modified HA (HA-CA) hydrogels have higher biocompatibility, better tissue adhesion properties, and provide improved stem cell survival and functionality compared to conventional HA hydrogels [16]. However, their potential use to treat DBW remains largely unexplored.

The purpose of this study was to assess the therapeutic effects of an HA-CA patch combined with ADSCs in the treatment of DBW. In particular, we evaluated their effects on tissue regeneration and angiogenesis using a DBW mouse model.

## 2. Results

### 2.1. Wound Area Measurement

The wounds were photographed on days 1, 3, 5, 7, 14, and 21, and the wound areas were quantified using ImageJ software. The change in wound size over time was calculated as the percentage of wound closure for each treatment group compared to the initial area of the wound and normalized relative to the initial area. Visual inspection indicated a decrease in wound size. Compared with control mice, the wound closure rate on day 3 was significantly higher in the treatment groups (control DBW group, 105% ± 18% vs. ADSCs group, 82% ± 15%, HA-CA group, 71% ± 11% **, HA-CA + ADSCs group, 76% ± 10% *; * *p* < 0.05, ** *p* < 0.01). At 14 days, remarkable wound healing was observed in the HA-CA + ADSCs group (control DBW group 59% ± 8% vs. HA-CA + ADSCs group 19% ± 4%; *p* < 0.01; Figure 1A,B). Although the wounds appeared to have healed in all groups on day 21, the wound size was significantly smaller in the HA-CA + ADSCs group than in the control and ADSCs groups (control DBW group, 16% ± 5% **, ADSCs group, 24% ± 17% * vs. HA-CA + ADSCs group, 8% ± 2%; * *p* < 0.05, ** *p* < 0.01; Figure 1A,B).

### 2.2. PKH26-Labeled ADSC Tracing

PKH26-labeled ADSCs (5 × 10^5^ cells/100 µL) were injected into healthy subcutaneous tissues at the wound boundary. In the HA-CA + ADSCs group, PKH26-labeled ADSCs were transplanted with the HA-CA patch at the wound site. The mice were sacrificed on day 14 for PKH26-labeled ADSC tracking. Although most ADSCs migrated from the injection site to the wound, only a few ADSCs were observed in the ADSC injection group. In contrast to the ADSC group, ADSCs were detected in the epidermis, papillary dermis, and reticular dermis at the wound site in the HA-CA + ADSCs group (Figure 1C).

### 2.3. Histopathological Assessment

Wounds were stained with hematoxylin and eosin on postoperative day (POD) 21 for histological observation. Complete re-epithelialization was observed in all groups; however, there were significant differences in epidermis thickness and skin appendages among the groups. In the DBW group, the epidermis consisted of one to three thin epithelial cell layers, and the extent of tissue regeneration at the dermis was low. In the ADSCs, HA-CA, and HA-CA + ADSCs groups, the epidermis was completely regenerated, and epidermal appendages, including sweat glands and hair follicles, were observed. Notably, neovascular-like structures were observed in the dermis layer (Figure 2A). Masson’s trichrome staining revealed marked granulation tissue formation in the HA-CA, ADSCs, and HA-CA + ADSCs groups. Furthermore, these groups showed increased proportions of muscle cells and extensive dermis tissue remodeling (Figure 2B). Terminal deoxynucleotidyl transferase dUTP nick end labeling (TUNEL)-positive cells were counted manually in microscopic fields (DBW group, 19 ± 2; ADSCs group, 25 ± 4; HA-CA group, 21 ± 4; HA-CA + ADSCs group, 21 ± 3; Figure 2F). The level of apoptosis was not significantly different among the groups (Figure 2C).

In comparison with the DBW group, the skin was significantly thicker in the HA-CA and HA-CA + ADSCs groups (DBW group, 468 ± 34; ADSCs group, 555 ± 52; HA-CA group, 815 ± 90 *; HA-CA + ADSCs group, 900 ± 102 μm *, * *p* < 0.05). The epidermis was thickest in the HA-CA + ADSCs group (DBW group, 77 ± 4; ADSCs group, 104 ± 16; HA-CA group, 107 ± 17; HA-CA + ADSCs group, 122 ± 7 μm *, * *p* < 0.05; Figure 2D,E).

### 2.4. Tissue Neovascularization

To examine wound neovascularization, we performed immunohistochemical staining for the endothelial cell-specific markers cluster of differentiation 31 (CD31) and von willebrand factor (vWF). Most of the CD31- and vWF-positive vessels were distributed in the lower middle part of the reticular layer of the dermis. CD31 is expressed in small vessels, mature blood vessels, and immature vascular structures. Most CD31-positive vessels were 10–15 μm in diameter and none exceeded 20 μm. Compared with the control DBW group, the number of CD31-positive vessels was higher in mice treated with HA-CA and HA-CA + ADSCs (control DBW group, 14 ± 2; ADSCs group, 25 ± 3; HA-CA group, 52 ± 5 *; HA-CA + ADSCs group, 73 ± 10 **; * *p* < 0.05, ** *p* < 0.01; Figure 3A,C). The vWF-positive vessels were larger than CD31-positive vessels and were 20–30 μm in diameter, with some exceeding 40 μm. Mice in the HA-CA + ADSCS group had the greatest number of vWF-positive vessels (control DBW group, 5 ± 1; ADSCs group, 6 ± 2; HA-CA group, 9 ± 1; HA-CA + ADSCs group, 16 ± 3 *; * *p* < 0.05; Figure 3B,D).

### 2.5. Angiogenesis-Related Gene Expression Profile

To confirm the effects of the different treatments on angiogenesis, we assessed the mRNA levels of various angiogenesis-associated genes, including those encoding vascular endothelial growth factor (VEGF), angiopoietin 1 (Ang-1), insulin-like growth factor 1 (IGF-1), fibroblast growth factor 2 (FGF-2), phosphoinositide 3-kinase (PI3K), protein kinase B (Akt), sprouty-related EVH1 domain-containing 1 (Spred1), and extracellular signal-regulated kinase (ERK). Mice treated with HA-CA + ADSCs had the highest *VEGF*, *IGF-1*, and *FGF-2* mRNA levels. *Ang-1* was upregulated in all treatment groups compared with the control group but was statistically significant only in the HA-CA + ADSCs group. *FGF-2* was expressed at higher levels in the HA-CA and HA-CA + ADSCs treatment groups than in the control group, but treatment with ADSCs alone had no effect on *FGF-2* mRNA levels. Although *PI3K* and *Akt* were significantly upregulated in mice treated with HA-CA + ADSCs, no changes were observed in the *Spred1* or *ERK* mRNA levels (Figure 4).

### 2.6. VEGF Expression Analysis by Immunofluorescence

To further confirm the effects of ADSCs and HA-CA on angiogenesis, we performed immunofluorescence staining for VEGF. Compared with the DBW group, in which the number of VEGF-expressing cells was low, the numbers of VEGF-positive cells were significantly higher in the ADSCs and HA-CA groups. In particular, mice treated with HA-CA + ADSCs had the greatest number of VEGF-expressing cells (Figure 4I). VEGF-positive cells were counted manually in microscopic fields and were highest in the HA-CA + ADSCs group (control DBW group, 15 ± 4; ADSCs group, 25 ± 6; HA-CA group, 28 ± 5; HA-CA + ADSCs group, 46 ± 5 **; ** *p* < 0.01; Figure 4J).

### 2.7. Angiogenesis-Related Protein Expression

We investigated the expression of various angiogenesis-associated proteins, including VEGFA, FGF-2, Akt, phospho-Akt, ERK1/2, and phospho-ERK1/2. VEGFA and FGF-2 expression was increased in the HA-CA and HA-CA + ADSCs treatment groups compared to the control DBW group (VEGFA, control DBW group, 49 ± 4; ADSCs group, 43 ± 14; HA-CA group, 70 ± 8; HA-CA + ADSCs group, 91 ± 2 *; * *p* < 0.05; Figure 5A). FGF-2 was increased in all groups compared to the controls, especially in the HA-CA group (control DBW group, 34 ± 8; ADSCs group, 54 ± 6; HA-CA group, 76 ± 3 **; HA-CA + ADSCs group, 52 ± 2; ** *p* < 0.01; Figure 5A). With regard to components of the ERK and PI3K signaling pathways, ERK1/2 and AKT1 expression was confirmed. Both the AKT and phospho-AKT (*p*-AKT) levels were increased in all treatment groups compared to the controls, and the difference was especially great in the HA-CA group (control DBW group, 20 ± 3; ADSCs group, 34 ± 5; HA-CA group, 56 ± 2 **; HA-CA + ADSCs group, 43 ± 2; ** *p* < 0.01; Figure 5B). The expression of total ERK1/2 was similar in all groups, and phospho-ERK1/2 (p-ERK1/2) was hardly expressed (control DBW group, 32 ± 3; ADSCs group, 28 ± 1; HA-CA group, 29 ± 2; HA-CA + ADSCs group, 27 ± 2; Figure 5B).

## 3. Discussion

Chronic hyperglycemia is the leading cause of vascular complications related to diabetes. Skin damage poses serious risks to diabetic patients due to the impaired healing abilities of the skin [17]. Diabetic foot ulcers are the most common complications in diabetic patients [18]. DBWs are caused by peripheral neuropathy, vascular dysfunction, and arterial occlusive disease due to persistent hyperglycemia [2,19]. Thus, limited joint mobility and foot deformities increase the risk of wounds and ulcers. In addition, hyperglycemia is linked to decreased white blood cell counts and macrophage function, and ischemic and neuropathic dysfunctions can lead to infections and delayed wound healing [18,20]. As the current DBW management methods have various limitations, the present study was performed to assess the potential clinical usefulness of the combination of biomaterials with stem cells.

Several biomaterials and stem cell types have been proposed for the treatment of DBW, and some of these have yielded encouraging results [21,22,23]. Biocompatibility, safety, degradability, and mechanical properties are crucial points to be taken into consideration when developing biomaterials intended for therapeutic use [24]. Woo et al. [22] assessed the effects of silk fibroin chitosan film and ADSCs in a DBW rat model and found that the biomaterial provided a biocompatible scaffold that could be used for stem cell delivery. Kanitkar et al. [25] reported that a polycaprolactone-gelatin nanofiber matrix exerted promising wound healing effects in a DBW mouse model.

Here, we investigated the effects of a HA-CA patch combined with ADSCs in a DBW mouse model. HA is a naturally occurring polysaccharide and a major component of the ECM. Due to its excellent viscoelastic properties and ability to promote cell migration, HA has emerged as a new therapeutic agent for use in regeneration and wound healing [26]. Many researchers are interested in the various functions of HA, and a number of types of biomaterials modified from HA have been developed. In addition, there is ongoing research in the field of wound medicine using various biomaterials [27,28]. In this study, 200 kDa HA was used and modified. To induce the crosslinking of HA to increase its retention time at the injury site, a mussel adhesion-inspired catechol group was introduced into the HA backbone. The HA-CA patch maximizes the effects of cell therapy by preventing stem cells from becoming attached to the scaffold and lost on transplantation. HA-CA patches provide excellent biocompatibility, tissue adhesion, and an improved survival rate, maximizing the regeneration ability of stem cells [16]. In this study, we showed that although both HA-CA and HA-CA + ADSCs reduced wound size, the wound healing effects of HA-CA + ADSCs were more potent. We believe that this synergistic effect reflects the fact that the HA-CA patch improved the regeneration ability of stem cells and maximized their paracrine effects.

The synergistic wound healing effects of the HA-CA patch and ADSCs were confirmed by histopathological analyses. The combination of HA-CA and ADSCs resulted in increased epithelialization, granulation tissue formation, and capillary formation. ADSCs alone or in combination with HA-CA promoted epidermis regeneration, neovascularization, and skin appendage development.

To confirm the effects of HA-CA and ADSCs on neovascularization in DBW, we stained tissues for CD31 and vWF. Although CD31-positive small vessels were observed in mice treated with ADSCs alone or in combination with HA-CA, the number of CD31-expressing vessels was significantly higher in the HA-CA + ADSCs group. Consistent with these observations, mice treated with HA-CA + ADSCs exhibited the greatest number of vWF-positive vessels. Although the therapeutic effects of stem cells in DBW have been reported previously [29], our findings suggest that ADSCs promote wound healing and neovascularization, and that their effects are augmented when combined with HA-CA.

ADSCs secrete various growth factors and cytokines, including VEGF, IGF-1, and FGF, promoting angiogenesis and tissue regeneration [16,30]. In this study, we showed that treatment with HA-CA combined with ADSCs significantly upregulated the expression of VEGF, Ang-1, IGF-1, and FGF-2. Importantly, mice treated with HA-CA + ADSCs exhibited the highest mRNA levels of *VEGF* and *IGF-1*. These results confirmed the synergistic neovascularization-promoting effects of HA-CA and ADSCs. Although the *PI3K* and *Akt* mRNA levels were significantly increased upon treatment with HA-CA + ADSCs, we found no changes in the expression levels of *ERK* and *Spred1*.

To confirm that the expression of mRNA related to angiogenesis was consistent at the protein level, proteins were extracted from the paraffin blocks, and VEGF, FGF-2, AKT-1, and ERK expression were confirmed. The VEGFA and FGF-2 expression levels were increased in the HA-CA and HA-CA + ADSCs groups. In particular, the expression of VEGFA showed the greatest increase in the HA-CA + ADSCs group and FGF-2 showed the greatest increase in expression in the HA-CA group. VEGFA is a potent inducer of angiogenesis and triggers most of the mechanisms involved in the activation and regulation of angiogenesis [31]. FGF-2 promotes epithelialization by mediating skin wound healing and strongly activates fibroblasts as well as other mesodermal-derived cells, including vascular endothelial and smooth muscle cells, osteoblasts, and chondrocytes. In addition, it is known to play a prevalent role in epidermal defect wound models [32]. This seems to be because wound healing progressed gradually with sacrifice of the animals at 3 weeks after HA-CA or ADSCs transplantation. In the HA-CA + ADSCs group, the progression of angiogenesis occurred, while in the HACA group, dermis cell proliferation seems to have been the main cause. Total ERK1/2 expression was confirmed in all groups, but there was little expression of p-ERK1/2 (activated ERK protein). Thus, angiogenesis was not mediated mainly through the mitogen-activated protein kinase (MAPK) pathway. The levels of AKT and p-AKT expression were increased in all treatment groups in comparison with the control DBW group, suggesting that angiogenesis was mediated through the PI3K/Akt pathway (Figure 6).

The PI3K/Akt pathway promotes endothelial cell proliferation, differentiation, and migration in response to tyrosine kinase growth factor receptors, including IGF-1 receptor, receptor tyrosine kinase receptor, and VEGFR [33,34]. Moreover, PI3K/Akt signaling increases Bcl-2 levels and decreases Bax levels, promoting cell survival [35]. The results of the present study indicate that the expression levels of various growth factors were increased in the HA-CA + ADSCs group compared with the ADSCs or HA-CA only treatment group. Therefore, by upregulating various growth factors and upstream pathway regulators, HA-CA and ADSCs activate the PI3K/Akt pathway promoting angiogenesis and tissue regeneration in DBW.

In conclusion, we investigated the effects of the HA-based biomaterial HA-CA and ADSCs in a mouse model of DBW. The combination of HA-CA and ADSCs showed synergistic wound healing effects via acceleration of tissue regeneration and angiogenesis. Additional clinical studies are warranted to confirm the clinical benefits of HA-CA combined with ADSCs in patients with DBW.

## 4. Materials and Methods

### 4.1. HA-CA Synthesis

The HA-CA patch was synthesized by conjugating dopamine hydrochloride (Sigma-Aldrich, St. Louis, MO, USA) to a 200 kDa HA backbone (Lifecore Biomedical, Chaska, MN, USA), as previously described [16,36]. Briefly, HA was dissolved in distilled water at a concentration of 1% (*w/v*). Subsequently, 1-ethyl-3-(3-dimethyl aminopropyl) carbodiimide (EDC; TCI Co., Japan) and N-hydroxysulfosuccinimide (NHS; Sigma-Aldrich) were added to HA solution at an HA:EDC:NHS molar ratio of 1:1.5:1.5 (pH 5.0). Dopamine hydrochloride was added to the solution at an HA:dopamine hydrochloride molar ratio of 1:1.5, and the solution was incubated for 12 h at room temperature. The solution was then dialyzed using a membrane (Cellu Sep T2, MW cut-off 6–8 kDa; Membrane Filtration Products Inc., Seguin, TX, USA) against pH 5.0 phosphate-buffered saline (PBS; Biosesang, Seongnam, Korea) to remove uncoupled dopamine hydrochloride. The resulting product was poured into Petri dishes in a thin layer and freeze-dried. The synthesis yield of HA-CA conjugate was about 80%. The substitution degree of catechol group to HA backbone was 8.8%, which was determined by measuring the absorbance of HA-CA solution at 280 nm wavelength using an ultraviolet-visible (UV-vis) light spectrophotometer (JASCO Corporation, Tokyo, Japan). To form a hydrogel, lyophilized HA-CA conjugate was dissolved in neutral PBS (Sigma) and mixed with oxidizing solution containing 4.5 mg/mL sodium periodate (NaIO4; Sigma) and 0.4 M sodium hydroxide (NaOH; Sigma).

### 4.2. ADSC Isolation and Characterization

Human ADSCs were isolated from three patients who underwent augmentation mammoplasty—the patients did not have inflammation or cancer. This study was approved by Seoul National University Bundang Hospital Institutional Review Board and Ethics Committee and conducted in accordance with the guidelines of the 1975 Declaration of Helsinki (IRB No. B-1702/381-301). Briefly, adipose tissue was washed with PBS and cut into smaller pieces. Enzymatic digestion was performed using 0.075% collagenase type I (Sigma-Aldrich) in a humidified 5% CO_2_ incubator for 1 h at 37 °C. After neutralization, samples were centrifuged, and supernatants were passed through a 100 µm cell strainer (BD Biosciences, Bedford, MA, USA). The cells were transferred into cell culture flasks with Dulbecco’s modified Eagle’s medium (Welgene, Gyeongsan, Korea) supplemented with 10% fetal bovine serum (Gibco, Carlsbad, CA, USA), 100 U/mL penicillin, and 100 μg/mL streptomycin (Lonza, Walkersville, MD, USA), and cells were maintained at 37 °C in a humidified 5% CO_2_ incubator. The ADSCs were used between passages 4 and 6 for fluorescence-activated cell sorting and animal experiments. ADSCs were characterized by flow cytometry for the cell-surface markers CD31, CD34, CD44, CD45, CD90, and HLA-DR (BD Biosciences Pharmingen, San Jose, CA, USA). To track the transplanted ADSCs, they were labeled with PKH26 red fluorescent dye (Sigma-Aldrich, St. Louis, MO, USA) according to the manufacturer’s instructions. Briefly, ADSCs were harvested and resuspended in 1 mL of diluent C solution. Then, 4 µL of PKH26 dye was added followed by incubation for 5 min. Fetal bovine serum (1 mL) was added for quenching for 2 min, followed by washing with PBS.

### 4.3. Diabetic Wound Animal Model

C57BL/6 male mice (7 weeks old, 23–26 g) were purchased from ORIENT BIO (Seongnam, Korea) and maintained according to the Association for Assessment and Accreditation of Laboratory Animal Care International system. All animal experiments conformed to the International Guide for the Care and Use of Laboratory Animals and were approved by the Institutional Animal Care and Use Committee of Seoul National University Bundang Hospital (IACUC No. BA1710-234/090-01).

Forty C57BL/6 mice were equally divided into four groups: control diabetic wound (DBW) group, ADSCs group, HA-CA group, and HA-CA + ADSCs group. Diabetes induction was performed by intraperitoneal injection of streptozotocin (STZ; Sigma-Aldrich) at a dose of 150 mg/kg dissolved in citrate buffer (pH 5.5) [37,38]. Blood was drawn from the tail vein, and the glucose level was determined using a glucometer (Accu-Check Performa; Roche, Pleasanton, CA, USA). The blood glucose level and body weights were measured every 3 days. Mice with blood glucose levels >250 mg/dL were considered diabetic. After 4 weeks of STZ administration, mice were anesthetized with 2% isoflurane inhalation. Excisional biopsy wounds on the shaved dorsal regions of the midline extending through the panniculus carnosus were made using a 6 mm punch. ADSCs (5 × 10^5^ cells/100 µL) labeled with PKH26 (Sigma-Aldrich) were transplanted into healthy tissue at the wound boundary, while HA-CA patches were injected at the wound site. The HA-CA patch (6 mm) was placed on the DBW (Figure 7). Wound areas were photographed on days 1, 3, 5, 7, 14, and 21 after ADSC and HA-CA transplantation. We identified the wound margins as whitish, dry, membrane-like structures, and measured the surface area using ImageJ software (version 1.51j8; National Institutes of Health, Bethesda, MD, USA). Changes in wound area over time were expressed as the percentage of the initial wound area.

### 4.4. Histopathological Assessment

DBW tissues were fixed in 10% formalin and embedded in paraffin. The tissues were routinely processed and cut into sections 4–5 μm thick. The sections were deparaffinized in xylene at room temperature and stained with hematoxylin and eosin (Cancer Diagnostics Inc., Durham, NC, USA) according to the manufacturer’s instructions. Masson’s trichrome staining (BBC Biochemical, Mount Vernon, WA, USA) was performed in accordance with the manufacturer’s protocol. Briefly, deparaffinized sections were fixed in Bouin’s solution for 1 h at 56 °C and stained with ClearView Iron Hematoxylin working solution for 10 min. Subsequently, tissues were stained with Biebrich scarlet-acid fuchsin solution (2 min), phosphomolybdic-phosphotungstic acid solution (10 min), aniline blue solution (3 min), and 1% acetic acid solution (2 min). ECM, collagen, and other connective tissue elements were stained blue and smooth muscles were stained red. DBW tissue sections were imaged (×100 magnification) using Carl Zeiss AxioVision 4 (Carl Zeiss MicroImaging GmbH, Jena, Germany).

### 4.5. Terminal Deoxynucleotidyl Transferase dUTP Nick End Labeling (TUNEL) Staining

In situ detection of apoptosis was performed by labeling DNA strand breaks in tissue sections using a TUNEL staining kit (Roche Diagnostics, Penzberg, Germany). Briefly, DBW tissue sections were pretreated with proteinase K (20 μg/mL) at 37 °C for 30 min and immediately washed. Subsequently, the DBW tissue sections were incubated with the TUNEL reaction mixture at 37 °C for 60 min. After mounting, sections were imaged under a fluorescence microscope (×200 magnification) using Carl Zeiss AxioVision 4 (Carl Zeiss MicroImaging GmbH).

### 4.6. Immunohistochemistry

Immunohistochemistry was performed using a GBI Polink-2 HRP kit (Golden Bridge International Inc., Bothell, WA, USA). Briefly, the sections were deparaffinized in xylene at room temperature and rehydrated in a graded series of ethanol. After heat-induced epitope retrieval in citrate buffer, pH 6.0 (Scytek Laboratories, Inc., West Logan, UT, USA), tissues were incubated in peroxidase blocking reagent for 15 min at room temperature. Subsequently, tissues were incubated with anti-von Willebrand factor (vWF; EMD Millipore, Temecula, CA, USA) and anti-CD31 (Thermo Fisher Scientific, Waltham, MA, USA) primary antibodies (1:50) for 90 min at room temperature, followed by incubation with diaminobenzidine chromogen. After dehydration, the sections were mounted with Histomount (National Diagnostics, Atlanta, GA, USA) and imaged (×400 magnification) using Carl Zeiss AxioVision 4 (Carl Zeiss MicroImaging GmbH).

### 4.7. RNA Isolation and qRT-PCR

RNA was isolated and purified using an RNeasy Plus Mini Kit (QIAGEN, Hilden, Germany) according to the manufacturer’s instructions. cDNA synthesis was performed using a High-Capacity cDNA Reverse Transcription Kit (Thermo Fisher Scientific). Primers for qRT-PCR were obtained from Cosmogenetech (Seoul, Korea), and the primer sequences are shown in Table 1. Reactions were prepared using Power SYBR Green PCR Master Mix (Applied Biosystems, Foster City, CA, USA) according to the manufacturer’s instructions, and were run on a ViiA 7 Real-Time PCR System (Life Technologies Corporation, Carlsbad, CA, USA) using the following cycling conditions: one cycle of denaturation at 95 °C/10 min, followed by 40 two-segment amplification cycles (95 °C/10 min, 60 °C/1 min). All reactions were performed in triplicate.

### 4.8. Immunofluorescence Analysis

Skin sections were deparaffinized in xylene and rehydrated in a graded ethanol series. After heat-induced epitope retrieval in citrate buffer, pH 6.0 (Scytek Laboratories, Inc.), sections were incubated with 3% bovine serum albumin blocking reagent for 10 min at room temperature. After blocking, sections were incubated with a primary antibody against VEGF (Santa Cruz Biotechnology, Inc., Santa Cruz, CA, USA) followed by incubation with Alexa Fluor 488 anti-rabbit secondary antibody (Biolegend, San Diego, CA, USA). The sections were mounted with 4′,6-diamidino-2-phenylindole (DAPI)-containing mounting medium (Vector Laboratories Inc., Burlingame, CA, USA) and observed under an inverted microscope (Axio Observer 7; Carl Zeiss Microscopy GmbH). VEGF-positive cells were counted manually in five microscopic fields in each stained sample, and the mean value was used for statistical analyses.

### 4.9. Protein Extraction and Western Blotting

Proteins were extracted from formalin-fixed paraffin-embedded (FFPE) samples using a Qproteome FFPE Tissue Kit (QIAGEN) according to the manufacturer’s instructions. The protein concentration was determined using Bio-Rad assay reagent (Bio-Rad, Hercules, CA, USA). Briefly, samples with equal concentrations of protein were mixed with 4× sample buffer (GenDEPOT Inc., Barker, TX, USA), heated at 95 °C for 10 min, and separated by 10% sodium dodecyl sulfate–polyacrylamide gel electrophoresis (SDS-PAGE). Proteins were then transferred onto polyvinylidene difluoride (PVDF) membranes (Amersham Life Science, Arlington Heights, IL, USA) in Tris-glycine transfer buffer (Invitrogen, Carlsbad, CA, USA). The membranes were blocked for 1 h at room temperature with 5% skim milk in Tris-buffered saline with Tween-20. The membranes were incubated at 4 °C overnight with anti-VEGFA (Abcam, Cambridge, UK), anti-FGF-2 (Santa Cruz Biotechnology, Inc.), anti-AKT1 (Santa Cruz Biotechnology, Inc.), anti-p-AKT1 (Santa Cruz Biotechnology, Inc.), anti-ERK1/2 (Santa Cruz Biotechnology, Inc.), anti-p-ERK (Santa Cruz Biotechnology, Inc.), or anti-β-actin (Santa Cruz Biotechnology, Inc.) primary antibodies, followed by incubation with horseradish peroxidase-conjugated anti-mouse IgG (Cell Signaling Technology, Danvers, MA, USA) or anti-rabbit IgG (Cell Signaling Technology) secondary antibodies as appropriate for 1 h at room temperature. The membranes were washed and then incubated using a West-Q Chemiluminescent Substrate Plus kit (GenDEPOT Inc.). The intensities of the protein bands were determined using Multi-Gauge software (version 3.0; Fuji Photo Film, Tokyo, Japan), and relative densities were expressed as ratios of control values. All reactions were performed in duplicate.

### 4.10. Statistical Analysis

Quantitative data were expressed as the mean ± standard deviation. Differences between groups were evaluated by one-way analysis of variance (ANOVA) followed by Dunn’s multiple comparison post hoc test. All statistical analyses were performed using PRISM v.5.01 (GraphPad Software, Inc., La Jolla, CA, USA) and *p* < 0.05 was taken to indicate statistical significance.

## Figures and Tables

**Figure 1 ijms-22-02632-f001:**
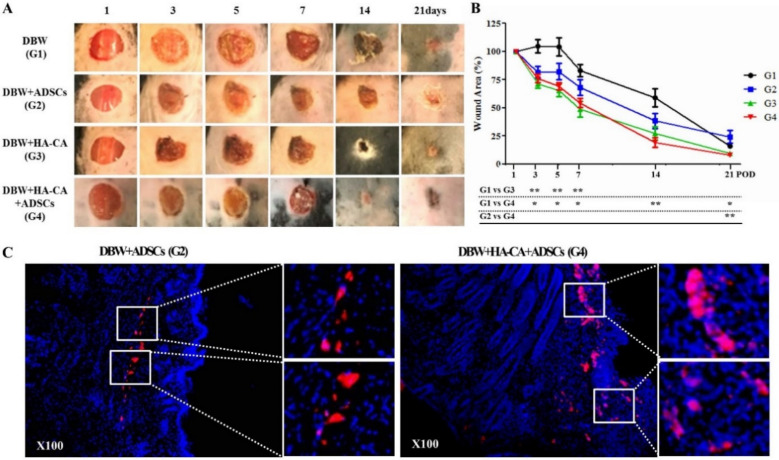
Effects of catechol-modified hyaluronic acid (HA-CA)/adipose-derived stem cells (ADSC) treatment on mouse diabetic wounds (DBW). (**A**) Wounds 6 mm in diameter were produced by punch biopsy, and the wound size was monitored on digital photographs using ImageJ on days 1, 3, 5, 7, 14, and 21. Compared with the DBW control group, the wound sizes were reduced from postoperative day (POD) 3 in mice treated with ADSCs or HA-CA. In particular, wound healing was accelerated after POD 7. The wounds of diabetic mice treated with HA-CA + ADSCs exhibited the fastest healing. (**B**) The fraction of the initial wound size was analyzed with ImageJ software. Compared with control mice, diabetic mice treated with HA-CA or HA-CA + ADSCs had significantly higher wound healing rates up to POD 7. After POD 7, HA-CA + ADSC-treated mice exhibited the fastest wound healing. The data are expressed as means ± standard error of the mean (SEM; * *p* < 0.05, ** *p* < 0.01; *n* = 8–10 mice per group). (**C**) ADSCs were stained with PKH26 red fluorescent dye to enable transplanted cell tracking. In contrast to the ADSC group, large numbers of PKH26-labeled ADSCs were detected in the epidermis, papillary dermis, and reticular dermis at the wound site in the HA-CA + ADSCs group (5 × 10^5^ cells/100 µL). Magnification: ×200.

**Figure 2 ijms-22-02632-f002:**
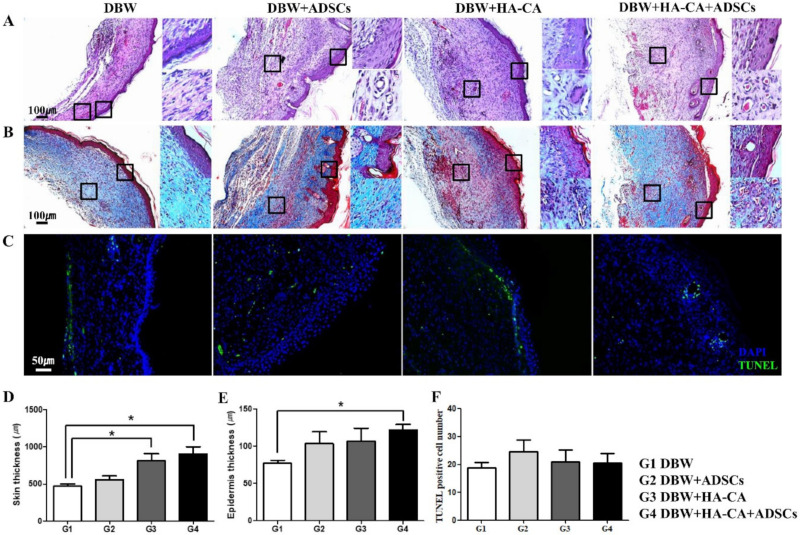
Histopathological analysis of the diabetic wounds. (**A**) Skin tissue sections were stained with hematoxylin and eosin (H&E) on POD 21. The epidermis was regenerated in all groups; however, the dermis and subcutis were not regenerated in the control DBW group. Blood vessels and epidermal appendages, including sweat glands and hair follicles, were observed in the HA-CA, ADSCs, and HA-CA + ADSCs groups. (**B**) Masson’s trichrome staining showing skeletal muscle (red) regeneration and collagen (blue) deposition in all groups. Compared with the control DBW group, the degree of skeletal muscle regeneration was higher in mice treated with HA-CA and/or ADSCs. (**C**) TUNEL staining was performed to assess apoptosis (green fluorescence). There were no differences in apoptosis among the groups. Magnification: ×100 (H&E, trichrome), ×200 (TUNEL). (**D**,**E**) Quantification of (**D**) skin thickness and (**E**) epidermis thickness. (**F**) Apoptotic cells were quantified. Data are presented as means ± SEM (* *p* < 0.05; *n* = 8–10 per group).

**Figure 3 ijms-22-02632-f003:**
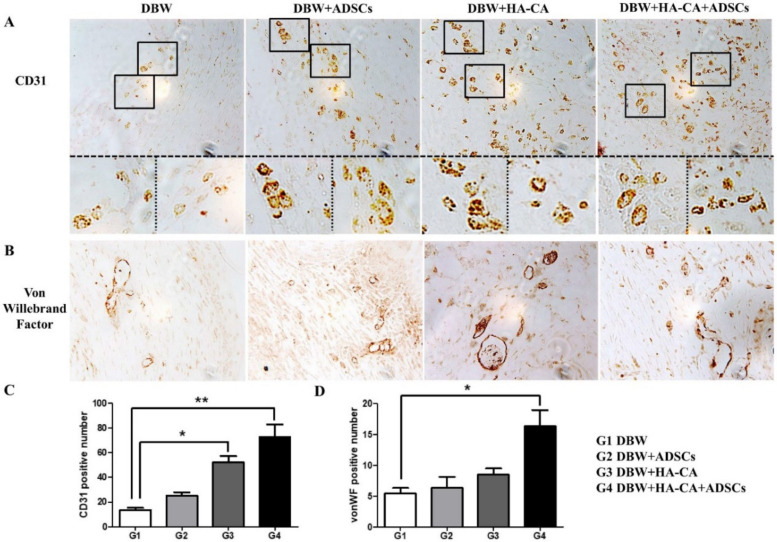
Immunohistochemical staining of the angiogenic markers cluster of differentiation 31 (CD31) and von willebrand factor (vWF). (**A**) Representative images of skin tissues showing positive staining for CD31 (brown) within the vessel structures in all groups. The number of CD31-positive vessels was higher in mice treated with HA-CA and HA-CA + ADSCs. (**B**) The number of vWF-positive vessels was extremely low in the control group, whereas mice in the HA-CA + ADSCs group had the greatest number of vWF-positive blood vessels. Magnification: ×400. (**C**,**D**) Bar graphs showing that mice in the HA-CA + ADSCs group had the greatest numbers of (**C**) CD31- and (**D**) vWF-positive vessels. Data are presented as means ± SEM (* *p* < 0.05, ** *p* < 0.01; *n* = 8–10 per group).

**Figure 4 ijms-22-02632-f004:**
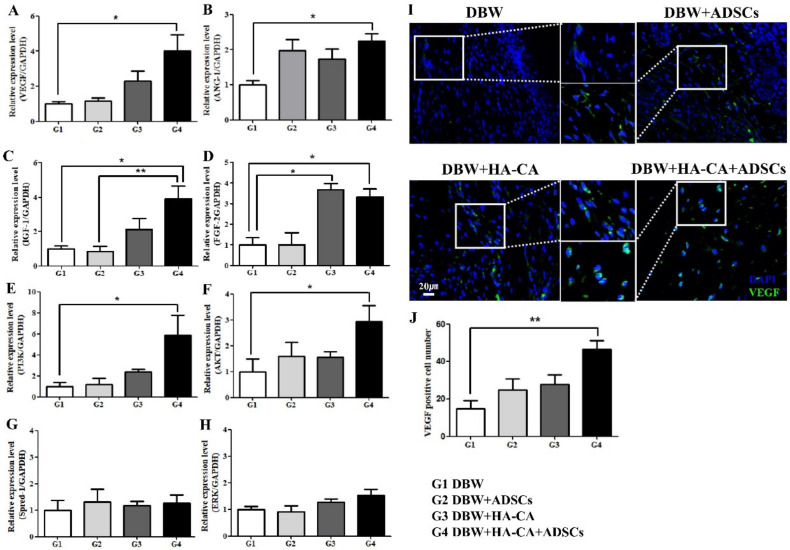
Effects of treatment with HA-CA and ADSCs on the angiogenesis-related gene expression profile. The mRNA levels of the angiogenesis-associated genes (**A**) *VEGF*, (**B**) *ANG-1*, (**C**) *IGF-1*, (**D**) *FGF-2*, (**E**) *PI3K*, (**F**) *AKT*, (**G**) *Spred1*, and (**H**) *ERK* were determined by quantitative real-time polymerase chain reaction (qRT-PCR). Data are presented as means ± SEM relative to *GAPDH* mRNA level (* *p* < 0.05, ** *p* < 0.01; *n* = 6 per group). (**I**) VEGF expression was assessed by immunofluorescence staining. VEGF expression was significantly increased in mice treated with HA-CA + ADSCs. (**J**) Bar graphs showing that mice in the HA-CA + ADSCs group had the greatest number of VEGF-positive cells (** *p* < 0.01, *n* = 8 per group). Magnification: ×400. VEGF, vascular endothelial growth factor; ANG-1, angiopoietin 1; IGF-1, insulin-like growth factor 1; FGF-2, fibroblast growth factor 2; PI3K, phosphoinositide 3-kinase; AKT, protein kinase B; Spred-1, sprouty-related EVH1 domain-containing 1; ERK, extracellular signal-regulated kinase; GAPDH, glyceraldehyde-3-phosphate dehydrogenase.

**Figure 5 ijms-22-02632-f005:**
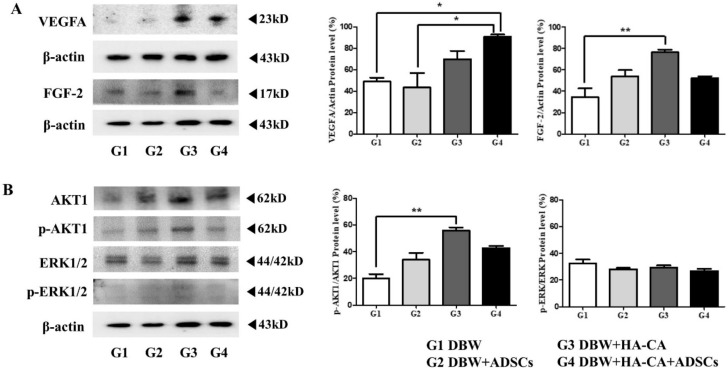
Angiogenesis-related protein expression after treatment with HA-CA and ADSCs. The expression levels of the angiogenesis-associated proteins (**A**) VEGFA and FGF-2, and (**B**) AKT1, p-AKT1, ERK1/2, and p-ERK1/2 were determined by Western blotting. Data are presented as means ± SEM relative to β-actin (* *p* < 0.05, ** *p* < 0.01; *n* = 4 per group). VEGFA, vascular endothelial growth factor A; FGF-2, fibroblast growth factor 2; AKT, protein kinase B; p-AKT, phospho-AKT; ERK1/2, extracellular signal-regulated kinase1/2; p-ERk1/2, phospho-ERK1/2.

**Figure 6 ijms-22-02632-f006:**
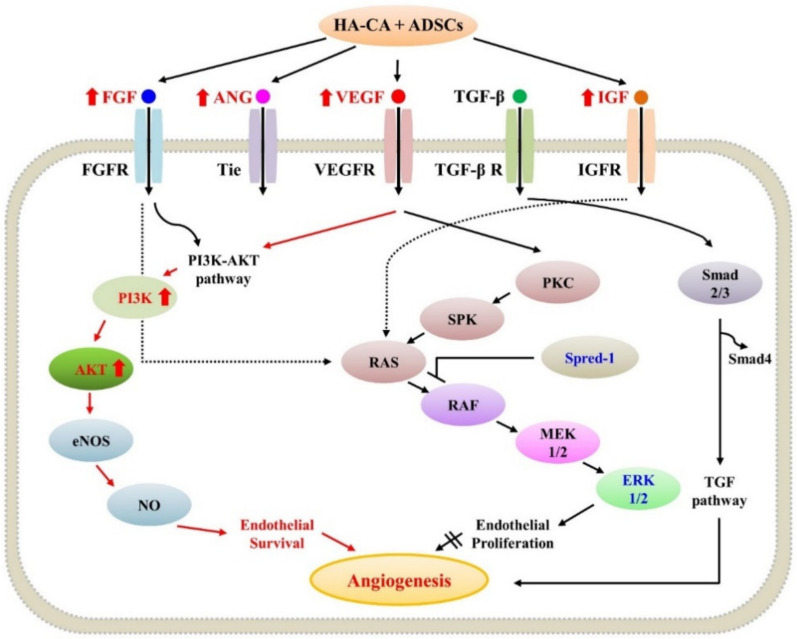
Schematic diagram of the angiogenesis pathway. FGF, ANG, VEGF, and IGF activate the PI3K/AKT pathway, promoting angiogenesis in the mouse diabetic wound model. FGF, fibroblast growth factor; VEGF, vascular endothelial growth factor; ANG, angiopoietin; IGF, insulin-like growth factor; PI3K, phosphoinositide 3-kinase; AKT, protein kinase B; Spred-1, sprouty-related EVH1 domain-containing 1; ERK, extracellular signal-regulated kinase.

**Figure 7 ijms-22-02632-f007:**
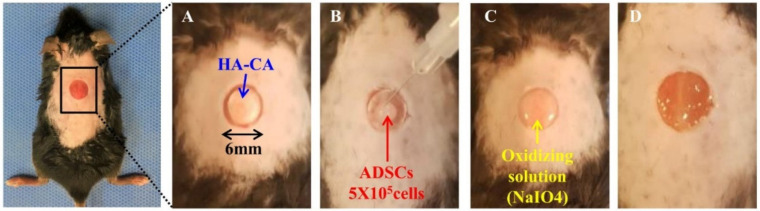
HA-CA and ADSC transplantation in the DBW mouse model. (**A**) HA-CA was transplanted into the 6 mm DBW, and (**B**) ADSCs (5 × 10^5^ cells) were deposited onto the HA-CA. (**C**) Oxidizing solution (NaIO_4_) was sprayed onto the cell-loaded HA-CA patch. (**D**) The crosslinked HA-CA patch was washed with phosphate buffered saline.

**Table 1 ijms-22-02632-t001:** Primer sequences.

Primer Name	Sequences
VEGF	F: 5′-GCA CAT AGA GAG AAT GAG CTT CC-3′ R: 5′-CTC CGC TCT GAA CAA GGC T-3′
ANG-1	F: 5′- ATC TTG ATA ACC GCA GCC AC-3′ R: 5′-TGT CGG CAC ATA CCT CTT GT-3′
IGF-1	F: 5′-ATG TAC TGT GCC CCA CTG AAG-3′ R: 5′-GTG TTT CGA TGT TTT GCA GGT-3′
FGF-2	F: 5′-GCT GGC TTC TAA GTG TGT-3′ R: 5′-CCA ACT GGA GTA TTT CCG TGA-3′
PI3K	F: 5′-CTC TCC TGT GCT GGC TAC TGT-3′ R: 5′-GCT CTC GGT TGA TTC CAA ACT-3′
AKT	F: 5′-ATG AAC GAC GTA GCC ATT GTG-3′ R: 5′-TTG TAG CCA ATA AAG GTG CCA T-3′
Spred-1	F: 5′-GAT GAG CGA GGA GAC GGC GAC-3′ R: 5′-GTC TCT GAG TCT CTC TCC ACG GA-3′
ERK1	F: 5′-GCG TTA CAT GTG GCA GCT TGA-3′ R: 5′-TGG AAC CCC ACC CCA TTT T-3′

Abbreviations: F, forward; R, reverse; VEGF, vascular endothelial growth factor, ANG-1, angiopoietin 1; IGF-1, insulin-like growth factor 1; FGF-2, fibroblast growth factor 2; PI3K, phosphoinositide 3-kinases; AKT, Protein Kinase B; Spred-1, sprouty-related EVH1 domain-containing 1; ERK, extracellular signal-regulated kinase.

## Data Availability

The data that support the findings of this study are available from the corresponding author upon reasonable request.

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
