# Peer review of "Effects of a Catechol-Functionalized Hyaluronic Acid Patch Combined with Human Adipose-Derived Stem Cells in Diabetic Wound Healing"

_ijms, 2021, doi:10.3390/ijms22052632_

Round 1

Reviewer 1 Report

This is a new revision from Pak et al. Overall, the manuscript is well-written.

The authors made efforts to address the previous comments and the manuscript is substantiated by improved data quantification/Figures. While not all new text appears to be highlighted, the authors have appropriately responded to all major issues.

Author Response

This is a new revision from Pak et al. Overall, the manuscript is well-written.

The authors made efforts to address the previous comments and the manuscript is substantiated by improved data quantification/Figures. While not all new text appears to be highlighted, the authors have appropriately responded to all major issues.

We appreciate the Reviewer 1’s positive comments.

Reviewer 2 Report

This manuscript describes the preparation and evaluation of dopamine-modified hyaluronic acid patches for use in diabetic wound healing. The paper could have an important scientific contribution; however, the material part must be significantly improved. Although the greater emphasis is placed on biological research, the material part must be described and presented in a reliable and accurate manner allowing others to repeat the presented research.

Specific comments:

  1. The authors' choice of the name "Catechol-Functionalized Hyaluronic Acid" instead of "Dophamine-Functionalized Hyaluronic Acid" is not very clear to me. In my opinion, the name should be more specific, even though dopamine undoubtedly contains the catechol moiety.
  2. Unfortunately, the dopamine-hyaluronic acid derivative has not been characterized in any way. The authors do not provide any evidence for a reaction between hyaluronic acid and dopamine (NMR or IR spectroscopy can be used for this purpose). There is no data on the degree of substitution (or dopamine content) in the sample (NMR or UV spectroscopy can be useful).
  3. The resulting material (patch) is also not characterized in any way. At the very least, the morphology of the sample should be described (by electron microscopy).
  4. Line 351: Provide here a methodology for cross-linking with sodium periodate.
  5. Across the text (lines 27, 95-103, 136-144, 165-170, 201-202, 218-230): The standard deviation should be expressed as ONE significant figure; that is, unless the number is between 11 and 19 times some power of ten, in which case you can use two significant figures. The mean value should be rounded off at the decimal place corresponding to the last significant digit of its standard deviation. E.g., 15.74% ± 4.94% (line 27) should be presented as 16 ± 5%; 24.23% ± 16.58% (line 27) should be presented as 24 ± 17%.

Round 2

Reviewer 2 Report

In the revised manuscript, the authors have successfully addressed all my concerns and have made the necessary revisions. The paper has been generally improved and, in my opinion, can be published in the present form.

This manuscript is a resubmission of an earlier submission. The following is a list of the peer review reports and author responses from that submission.

Round 1

Reviewer 1 Report

This is an interesting work for diabetic wound approaches using mesenchymal stem cells and hyaluronic acid (HA) delivery to the wound site. The study is concise and clear but authors should highlight the novelty in the proposed approach as other works have used HA and stem cells to improve wound regeneration. The conclusions are ambitious considering the results achieved. The manuscript requires additional revisions before publishing.

The introduction of hyaluronic acid is too general as there are HA molecules with different biological responses. Please include information on the HA used (low vs high molecular weight) etc… What was the rational for HA-CA modification? Why 200 kDa HA?

How was the deep of the wound calculated?

The scale bars are missing in all images.

Did authors measure the diameter of the formed vessels? What were the parameters to define small and large vessels? Were the vessels more at the surface or deeper into the skin?

Discussion:

Please relate the outcomes of this study with the current gold standards for DBW.

“Both patches and ADSCs have been previously shown to promote wound healing” Please clarify what is the novelty of the work if HA and ADSCs have already been approached for wound healing.

“…HA-CA improved the survival of stem cells…”. How did authors assess the viability and/or survival of implanted ADSCs? Figure 2 shows that cells are present at the site of interest but cell labelling does not necessarily mean cells are viable.

Authors did not measure secreted factors by ADSCs and the gene expression is not sufficient to prove cells are expressing growth factors. Thus, the scheme on the angiogenic pathway is too ambitious for the results shown, especially when it is not proven cells are alive after implantation. Authors should present data on protein expression and use blocking molecules/agonists to confirm the intracellular pathways.

Figures

Figure 1 – Please indicate what G2 and G3 mean.

Figure 2 – The quality of histological images is not good. Please improve image resolution.  Please indicate what is represented in green and in Blue (fluorescence) in Figure 2.

Figure 4 – It is quite interesting that only ANG-1 in G4 is statistically significant to control conditions. Please comment on that.

Author Response

Reviewer 1

Comments and Suggestions for Authors

This is an interesting work for diabetic wound approaches using mesenchymal stem cells and hyaluronic acid (HA) delivery to the wound site. The study is concise and clear but authors should highlight the novelty in the proposed approach as other works have used HA and stem cells to improve wound regeneration. The conclusions are ambitious considering the results achieved. The manuscript requires additional revisions before publishing.

Thank you for the opportunity to revise and resubmit our manuscript. In this letter, we respond to each of the reviewers’ comments with our response (in blue).

The introduction of hyaluronic acid is too general as there are HA molecules with different biological responses. Please include information on the HA used (low vs high molecular weight) etc… What was the rational for HA-CA modification? Why 200 kDa HA?

- We appreciate the reviewer for this valuable comment. As the reviewer pointed out, it has been reported that the molecular size of HA influences its biological activities (Int. J. Cell Biol. 2015;2015:563818). Generally, it is known that HA with high molecular weight inhibits proliferation and migration of most cells, while low molecular weight forms of HA (< 300 kDa) promotes cell proliferation and displays angiogenic properties (Int. J. Cancer 1997;71:251–256, J. Biol. Chem. 2002;277:41046–41059, Int. J. Mol. Sci. 2019;20(15):3722.). Especially, HA with 150–250 kDa size range has shown a beneficial effect on wound healing by enhancing cell-HA interactions through cell surface receptors for HA (CD44 and RHAMM), which activates signal transduction essential for cellular migration and proliferation (Wound Repair Regen. 2014;22(4):521–526, Glycobiology 2017;27(9):868–877]. Based on these findings, 200 kDa HA was used in this study. For inducing the crosslinking of HA to increase its retention time at the injury site, a mussel adhesion-inspired catechol group was introduced to the HA backbone. The catechol-functionalized HA (HA-CA) hydrogel has been proved its biocompatibility, tissue-adhesiveness, and regenerative efficacy of stem cell/drug delivery in various types of animal defect models (Adv. Funct. Mater. 2015;25(25):3814-3824, Biomacromoleulces 2016;17(6):1939-1948, Nat. Commun. 2018;9:5402). This is why we chose the HA-CA patch as a matrix for DBW treatment. We have now mentioned why we used 200kDa HA-CA hydrogel for this study in the Introduction section.

How was the deep of the wound calculated?

- Changes in wound areas over time were expressed by the percentage of the initial wound areas. We have included this information in the Material and Methods section.

We did not consider the depth of the wound because the surgery of the diabetic wound animal model was performed in the same way. Only skins of the same size (6 mm) were removed.

The scale bars are missing in all images.

- We corrected according to the reviewer's comment. The scale bars were displayed in all Figures.

Did authors measure the diameter of the formed vessels? What were the parameters to define small and large vessels? Were the vessels more at the surface or deeper into the skin?

- Most CD31-positive vessels are 10-15㎛ in diameter and do not exceed 20㎛. On the other hand, vWF-positive vessels were 20-30㎛ in diameter, and some vessels were more than 40㎛. In the manuscript, the vessels < 20㎛ were classified as small, and vessels > 20㎛ were classified as relatively large. We performed immunohistochemistry to identify the CD-31 and vWF expressions in each. If we had double-stained CD31 and vWF through immunofluorescence staining, we could have more reliably confirmed the expression patterns of both markers in blood vessels. Most of the CD31- and vWF-positive vessels were distributed in the lower middle of the dermis’s reticular layer. We have now mentioned this in the Manuscript.

Discussion:

Please relate the outcomes of this study with the current gold standards for DBW.

“Both patches and ADSCs have been previously shown to promote wound healing” Please clarify what is the novelty of the work if HA and ADSCs have already been approached for wound healing.

- There was an error in the sentence expression, so I corrected it.

Many researchers are interested in the various functions of hyaluronic acid, and various types of biomaterials modified from hyaluronic acid have been published. Also, wound medicine using diverse biomaterials are continuously being studied (Int J Biol Macromol. 2020 Jul 16;164:667-676. / Tissue Eng Regen Med. 2020 Jul 16.)

We adopted a diabetic wound animal model to evaluate the Catechol-Functionalized Hyaluronic Acid. The diabetic wound is difficult to treat as a general wound treatment. In this study, hyaluronic Acid was modified using Catechol to improve the function of biomaterials, which was used as a patch.

The patch type maximizes the effect of cell therapy by preventing stem cells from being attached to the scaffold and being lost to the outside when stem cells are transplanted. We have now mentioned this in the Discussion section.

“…HA-CA improved the survival of stem cells…”. How did authors assess the viability and/or survival of implanted ADSCs? Figure 2 shows that cells are present at the site of interest but cell labelling does not necessarily mean cells are viable.

- There was an error in the sentence expression, so I corrected it.

HA-CA patch improved the regeneration ability of stem cells and maximized their paracrine effects.

Authors did not measure secreted factors by ADSCs and the gene expression is not sufficient to prove cells are expressing growth factors. Thus, the scheme on the angiogenic pathway is too ambitious for the results shown, especially when it is not proven cells are alive after implantation. Authors should present data on protein expression and use blocking molecules/agonists to confirm the intracellular pathways.

- Since many research already found that the paracrine effects of growth factors secreted from ADSCs, we did not directly perform this in this study. Previous studies have shown that ADSCs secrete various growth factors and cytokines, including VEGF, IGF-1, and FGF, and promotion of angiogenesis and tissue regeneration.

Our results show that the expression of various growth factors was increased in the HA-CA+ADSCs group compared with ADSCs or HA-CA only treatment groups. If a large number of ADSCs were not present (survived) in the HA-CA+ADSCs group, the expression of growth factors should be similar to those of the ADSCs or HA-CA group. We have now mentioned this in the Discussion session.

We agree with the reviewer's comment that both mRNA and protein expression should be verified.  However, we have not obtained enough animal tissues to confirm protein expression. Since the Association for Assessment and Accreditation of Laboratory Animal Care International system recommends extremely strict and minimal animal sacrifices, we had a limit on the number of animals. Our wound size was very small (6mm diameter), and half of them were used for Histopathological assessment and the other half for RNA expression. Since the protein expression could not be confirmed, the expression of VEGF was confirmed through immunofluorescent staining. This was consistent with the mRNA expression pattern. We are well aware that this result cannot be interpreted that RNA and protein expression are consistent. We are looking forward a future research to identify both protein and RNA expression.

Figures

Figure 1 – Please indicate what G2 and G3 mean.

- We corrected the Figure 1 according to the reviewer's comment. G2 and G3 mean were represented in the Figure.

Figure 2 – The quality of histological images is not good. Please improve image resolution.  Please indicate what is represented in green and in Blue (fluorescence) in Figure 2.

- The image resolution has been changed from 300dpi to 600dpi. (The journal system does not attach each Figure file but uploads it by inserting it into the MS word file.)

- We corrected the Figure 2 according to the reviewer's comment. Green and blue fluorescence were represented in the Figure.

Figure 4 – It is quite interesting that only ANG-1 in G4 is statistically significant to control conditions. Please comment on that.

- We conducted statistical analysis using one-way analysis of variance (ANOVA) followed by Dunn’s multiple comparison post-hoc tests. We agreed with the reviewer's comment. We analyzed G1 and other groups through t-test, respectively, and found that they had statistically significant group difference. However, the t-test was not appropriate for analyzing our results. G4 has a statistical significance because the mean value is slightly larger than that of G2 and G3 and the standard deviation is relatively small.

Reviewer 2 Report

> This is an elegant study with potentially important observations for
> the field. There are however some minor editorial issues which
> require address and a single major issue of interpretation;
> subjective vs. objective. The later issue runs throughout the
> manuscript.
>
> Abstract.
>
> DBW requires description before abbreviation.
>
> Ln20. Diabete?
>
> Ln21. Needs to be explicitly detailed what the delivery of ADSC 'at'
> the patch entails. Is this on top of, into, underneath, at the edges
> of, or some other location. This should also be clearly highlighted
> within Results and M&M.
>
> Ln25. This is the first interpretative instance. Looking through the
> manuscript it is fairly clear that HA-CA supports wound closure while
> ADSC drive angiogenesis. i.e. HA-CA and HA-CA+ADSCs promote wound
> closure and not just HA-CA+ADSCs as stated.
>
> Introduction.
>
> No comments.
>
> Results.
>
> Ln76. A brief intro here would be useful. i.e. what wounds and what
> has been done to them?
>
> Ln83-86. G3 and G4 are comparable but G3 does not get mentioned? No
> sig diff between G3 and G4 indicates they are the same.
>
> Ln.92. This statement does not reflect the data presented.
> HA-CA+ADSC is not faster than HA-CA.
>
> Ln 95-96. Is this true. G4 is more rapid between Dy 7 and Dy 14 but
> equally G3 faster between Dy14 and Dy 21. Both end up at the same
> place at Day 21 so how can one be faster than the other?
>
> Ln104. As highlighted above. Greater detail required around ADSC
> delivery when combined with HA-CA patch.
>
> Ln123. HA-CA vs. HA-CA+ADSC = comparable skin thickness. Therefore
> HA-CA is driving accelerated wound closure.
>
> Ln143 + Ln146. HA-CA+ADSCs have elevated CD31 and vWF = ADSC are
> promoting angiogenesis.
>
> Discussion.
>
> Ln211, Ln214-218. These sections need to be rephrased to reflect
> above i.e. ADSCs promote angiogenesis, HA-CA drives repair. You need
> both for an effective vascularised repair tissue.

Author Response

This is an elegant study with potentially important observations for the field. There are however some minor editorial issues which require address and a single major issue of interpretation; subjective vs. objective. The later issue runs throughout the manuscript.

Thank you for the opportunity to revise and resubmit our manuscript. In this letter, we respond to each of the reviewers’ comments with our response (in blue).

Abstract.

DBW requires description before abbreviation.

- We corrected according to the reviewer's comment

Ln20. Diabete?

- We corrected according to the reviewer's comment

Ln21. Needs to be explicitly detailed what the delivery of ADSC 'at' the patch entails. Is this on top of, into, underneath, at the edges of, or some other location. This should also be clearly highlighted within Results and M&M.

- hADSCs were evenly transplanted throughout the patch (6 mm diameter). We have now mentioned this in the Material and Methods section.

Ln25. This is the first interpretative instance. Looking through the manuscript it is fairly clear that HA-CA supports wound closure while ADSC drive angiogenesis. i.e. HA-CA and HA-CA+ADSCs promote wound closure and not just HA-CA+ADSCs as stated.

- We appreciate the reviewer for this valuable comment. HA with high molecular weight inhibits proliferation and migration of most cells, while low molecular weight forms of HA (< 300 kDa) promotes cell proliferation and displays angiogenic properties (Int. J. Cancer 1997;71:251–256, J. Biol. Chem. 2002;277:41046–41059, Int. J. Mol. Sci. 2019;20(15):3722.). Especially, HA with 150–250 kDa size range has shown a beneficial effect on wound healing by enhancing cell-HA interactions through cell surface receptors for HA, which activates signal transduction essential for cellular migration and proliferation (Wound Repair Regen. 2014;22(4):521–526, Glycobiology 2017;27(9):868–877]. Based on these findings, 200 kDa HA was used in this study. For inducing the crosslinking of HA to increase its retention time at the injury site, a mussel adhesion-inspired catechol group was introduced to the HA backbone.

Since many research already found that the paracrine effects of growth factors secreted from ADSCs, we did not directly perform this in this study. Previous studies have shown that ADSCs secrete various growth factors and cytokines, including VEGF, IGF-1, and FGF, and promotion of angiogenesis and tissue regeneration. Therefore, the combination of HA-CA and hADSCs showed synergistic wound healing effects, maximizing the paracrine effects and survival of hADSCs. We have now mentioned this in the Manuscript.

Introduction.

No comments.

Results.

Ln76. A brief intro here would be useful. i.e. what wounds and what has been done to them?

- We have now added detailed explanations.

> Ln83-86. G3 and G4 are comparable but G3 does not get mentioned? No

> sig diff between G3 and G4 indicates they are the same.

> Ln.92. This statement does not reflect the data presented.

> HA-CA+ADSC is not faster than HA-CA.

> Ln 95-96. Is this true. G4 is more rapid between Dy 7 and Dy 14 but equally G3 faster between Dy14 and Dy 21. Both end up at the same place at Day 21 so how can one be faster than the other?

- (2.1 Wound area measurement section) We conducted statistical analysis using one-way analysis of variance (ANOVA) followed by Dunn’s multiple comparison post-hoc tests. Also, we compared all pair of columns in each timepoint. On day 3, the wound closure rate was significantly higher in the treatment groups compared with control (G1). On day 7, the wound size was decreased in G3 and G4 compared with control (G1, 83.42 ± 5.35% vs. G2, 66.24 ± 7.10%, G3, 48.74 ± 7.16%**, G4, 54.36 ± 4.00%*, *P < 0.05, **< 0.01). At 14 days, we observed remarkable wound healing in the HA-CA+hADSCs group (G1, 58.80 ± 8.04% vs. G2, 38.31 ± 6.83%, G3, 27.39 ± 5.94%, G4, 19.21 ± 4.36%**; **P < 0.01). On day 21, mice in the HA-CA+hADSCs group exhibited a significantly smaller wound size compared with mice in the control and hADSCs groups (G1, 15.74 ± 4.94%**, G2, 24.23 ± 16.58%*, G3, 9.65 ± 1.88% vs. G4, 7.69 ± 2.12%; *P < 0.05, **< 0.01). However, HA-CA group on day 14 and 21 has not a statistical significance. We have now added this information in the Results section.

> Ln104. As highlighted above. Greater detail required around ADSC delivery when combined with HA-CA patch.

- We mentioned ADSC delivery at HA-CA patch in the Material and Methods section. We have now added explanations according to the reviewer's comment.

hADSCs (5 × 105 cells/100 µL) labeled with PKH26 were transplanted into healthy tissue at the wound boundary, while HA-CA patches were injected at the wound site. HA-CA was transplanted into the 6-mm DBW, and hADSCs were evenly transplanted throughout the HA-CA patch. Oxidizing solution (NaIO4) was sprayed onto the cell-loaded HA-CA patches. The crosslinked HA-CA patch was washed with PBS.

Ln123. HA-CA vs. HA-CA+ADSC = comparable skin thickness. Therefore HA-CA is driving accelerated wound closure.

Ln143 + Ln146. HA-CA+ADSCs have elevated CD31 and vWF = ADSC are promoting angiogenesis.

Discussion.

Ln211, Ln214-218. These sections need to be rephrased to reflect above i.e. ADSCs promote angiogenesis, HA-CA drives repair. You need both for an effective vascularised repair tissue.

- We agree with the reviewer's comment and have now added this information in Discussion section.

The catechol-functionalized HA (HA-CA) hydrogel has been proved its biocompatibility, tissue-adhesiveness, and regenerative efficacy of stem cell/drug delivery in various types of animal defect models (Adv. Funct. Mater. 2015;25(25):3814-3824, Biomacromoleulces 2016;17(6):1939-1948, Nat. Commun. 2018;9:5402). In this study, we investigated the effects of HA-CA patch type combined with hADSCs.

Most of the CD31- and vWF-positive vessels were distributed in the lower middle of the dermis’s reticular layer. CD31 is expressed in small vessels, mature blood vessels, and immature vascular structures. Most CD31-positive vessels are 10-15㎛ in diameter and do not exceed 20㎛. Compared with the control group, the number of CD31-positive vessels was higher in mice treated with HA-CA and HA-CA+hADSCs. The vWF-positive vessels were 20-30㎛ in diameter, and some vessels were more than 40㎛. The vessels relatively larger than CD-positive vessels. Mice in the HA-CA+ADSCS group had the highest number of vWF-positive vessels.

Reviewer 3 Report

The manuscript is well-written. The authors propose to study diabetic wound healing using a mouse model treated with streptozotocin and delivery of hyaluronic acid patch, human adipose progenitors or a combination of both. Major issues are that the specific relevance of the model for diabetic wound healing is not clearly demonstrated or supported and that survival of adipose progenitors is not well established.

The authors should specify in title and/or abstract that they study “human” ADSC.

Abstract

Define DBW at first use.

Introduction

Clarify the statement: “MSC are more stable than other stem cell types”

Methods

The site of origin of ADSC is not clear. The authors only specify that the patients were undergoing augmentation mammoplasty. Please specify.

The authors do not indicate if the ADSC were expanded for multiple passages and at what passage they were characterized by FACS and used in this study. Please also confirm the immunophenotype (CD31, CD34, CD45, CD73, CD105, and HLA-DR).

Results

The authors utilize STZ treated B/6 mice with punched wounds to model DBW. The control group is STZ-treated mice with wound recovery at 21 days. In a previous study, the rate of wound healing has been reported to be the same as non-STZ treated animals (Nishikori Y, Shiota N, Okunishi H. The role of mast cells in cutaneous wound healing in streptozotocin-induced diabetic mice. Arch Dermatol Res 2014;306:823–835), with reported differences in vascularization and mast cells.

The authors need to discuss the choice of this model for DBW over other models (e.g., rat,…). The authors need to provide either references or data supporting the relevance of this model to study DBW specifically compared with general (non-diabetic) wound healing studies. This may be presented by providing control data from non-STZ-treated controls. The authors should document local angiogenesis in controls and local immune cells (e.g., mast cells, lymphocytes and/or macrophages).

Also, the authors are delivering human ADSC in B/6 mice without immunosuppressant. The authors should document local immunogenicity and immune cells at the site of wound in their paraffin blocks.

The sole detection of PKH26 does not necessarily support survival of injected ADSC. The authors should co-stain the area with at least one human specific antigen (e.g., human nuclear antigen, human nucleolar antigen, human laminA/C, human mitochondria,…). Because ADSC may not survive, it is not clear if the authors could possibly get the same results using ADSC-conditioned medium.

At what timepoint, is PKH26 detected in the tissue (Figure 1C)?

Please define exactly what tissue and how many replicates are analyzed by RT-PCR. A non-STZ treated control would also have been relevant.

Please provide adequate quantification (and specify methodology) for Figure 4I to support that “VEGF expression was significantly increased” and for TUNEL assay in Figure 2C (i.e., Quantitative analysis of multiple sections throughout the wounded area using low magnification or stitched images).

Please specify what n=8-10 means in Figure legends 2 and 3 (how many sections per wound/animal how were they spaced within the tissue, how many animals?).

When the authors report magnifications at x400 or x100, do they include only the objectives or is this using the oculars?

Please specify “Compared with the control (DBW) group, skeletal muscle regeneration was higher. How did you quantify the Masson’s trichrome staining? The authors should document a non-wounded area as control adjacent to the wound in their current sections or in another animal.

Author Response

Reviewer 3

 The manuscript is well-written. The authors propose to study diabetic wound healing using a mouse model treated with streptozotocin and delivery of hyaluronic acid patch, human adipose progenitors or a combination of both. Major issues are that the specific relevance of the model for diabetic wound healing is not clearly demonstrated or supported and that survival of adipose progenitors is not well established.

Thank you for the opportunity to revise and resubmit our manuscript. In this letter, we respond to each of the reviewers’ comments with our response (in blue). We were given a 10-day revision period from the editor, so we couldn't perform any additional experiments suggested by the reviewer. However, data analysis was done faithfully according to the comments.

The authors should specify in title and/or abstract that they study “human” ADSC.

 - We appreciate the reviewer for this valuable comment, and have now added the “human” in the title and abstract

Abstract

Define DBW at first use.

- We added the definition of DBW at the first place according to the reviewer's comment.

Introduction

Clarify the statement: “MSC are more stable than other stem cell types”

 - We appreciate the Reviewer’s comment and have now added detailed explanations in the Introduction section as follows:

Because mesenchymal stem cells (MSCs) are more accessible and safer than other stem cell types, many clinical trials are in under development. MSCs are stable in supply for experiments or clinical treatment because of obtaining from various tissues (Biosci Rep. 2015 Apr 28;35(2):e00191). We have now added this information in the Introduction section.

Methods

The site of origin of ADSC is not clear. The authors only specify that the patients were undergoing augmentation mammoplasty. Please specify.

- Augmentation mammoplasty is a breast augmentation procedure that removes adipose tissue to place an implant in the breast. We performed ADSCs primary culture with adipose tissue that was discarded. This study was approved by the Institutional Review Board and Ethics Committee and conducted in accordance with the guidelines.

The authors do not indicate if the ADSC were expanded for multiple passages and at what passage they were characterized by FACS and used in this study. Please also confirm the immunophenotype (CD31, CD34, CD45, CD73, CD105, and HLA-DR).

- The ADSCs were used between 4-6 passages for FACS and animal experiments. We have now mentioned this content in the Material and Methods section.

Results

The authors utilize STZ treated B/6 mice with punched wounds to model DBW. The control group is STZ-treated mice with wound recovery at 21 days. In a previous study, the rate of wound healing has been reported to be the same as non-STZ treated animals (Nishikori Y, Shiota N, Okunishi H. The role of mast cells in cutaneous wound healing in streptozotocin-induced diabetic mice. Arch Dermatol Res 2014;306:823–835), with reported differences in vascularization and mast cells.

The authors need to discuss the choice of this model for DBW over other models (e.g., rat,…). The authors need to provide either references or data supporting the relevance of this model to study DBW specifically compared with general (non-diabetic) wound healing studies. This may be presented by providing control data from non-STZ-treated controls. The authors should document local angiogenesis in controls and local immune cells (e.g., mast cells, lymphocytes and/or macrophages).

- We appreciate the Reviewer’s valuable comment. We conducted a systematic review of a diabetic wound animal model and chose STZ 150mg/kg treatment. In many previously published papers, diabetes was induced with STZ, and wound healing was delayed compared to normal (non-STZ treatment) mice. Induction of diabetic mice with STZ is widely known and well accepted. These References were added in the Material and Methods session. According to Lim et al (2015), diabetic mice were induced by a single intraperitoneal injection of streptozotocin (150 mg per kg body weight) as our study. As shown in the picture below, the wound was recovered 10 days in the normal mouse, and the diabetic mouse induced with STZ 150mg/kg completely recovered on the 20th day (J Invest Dermatol. 2015 Jan;135(1):269-278.).

According to Long et al., the mice were intraperitoneally injected with STZ (50 mg/kg, pH 4.5) dissolved in sodium citrate or injected with vehicle (sodium citrate only) for five consecutive days. Also, diabetic wound healing was delayed in the STZ treatment group compared with normal control (Diabetes 2016 Mar; 65(3): 780-793.).

Deeds et al. demonstrated Single Dose Streptozotocin Induced Diabetes. They evaluated the diabetic mice models with various doses of STZ in various strain mice (Lab Anim. 2011 Jul; 45(3): 131–140.).

I reviewed Nishikori's paper according to the reviewer's comment, and it was very interesting. This is valuable information. We will consider reviewers' comments such as non-STZ treatment group and non-wounded area in further study.

Also, the authors are delivering human ADSC in B/6 mice without immunosuppressant. The authors should document local immunogenicity and immune cells at the site of wound in their paraffin blocks.

- The immunomodulatory properties of mesenchymal stem cells are known to escape immune recognition and even actively inhibit immune responses. So we transplanted human adipose-derived stem cells into a diabetic wound mouse model and confirmed the therapeutic effect.

One of the reasons for the attention of MSC in the induction of immunological tolerance is that it does not induce an immune response even after xenotransplantation. (Stem Cells Transl Med. 2012 Mar; 1(3): 200–205.)

This low immunity of MSC is thought that MSC expresses type 1 histocompatibility antigen but does not express type 2 histocompatibility antigen. In other words, when MSCs are transplanted between allotypes and xenotypes, MSCs express type 1 histocompatibility antigens to protect MSCs from spontaneous killer cell-mediated cell destruction of the host.However, MSCs do not express type 2 histocompatibility antigens. MSCs can be avoided by transplantation with effector CD4 T cells. In addition, not expressing Fas ligand and costimulatory factors (B7-1, B7-2, or CD40) in MSC is thought to induce low immunity of MSC.

The sole detection of PKH26 does not necessarily support survival of injected ADSC. The authors should co-stain the area with at least one human specific antigen (e.g., human nuclear antigen, human nucleolar antigen, human laminA/C, human mitochondria,…). Because ADSC may not survive, it is not clear if the authors could possibly get the same results using ADSC-conditioned medium.

At what timepoint, is PKH26 detected in the tissue (Figure 1C)?

- I agree with the reviewer's comment, so I addressed this in the manuscript.

“HA-CA patch improved the regeneration ability of stem cells and maximized their paracrine effects.”

PKH26-labeled hADSCs do not support the survival of hADSCs. It can confirm whether the transplanted hADCS is present in the tissue or how it is distributed. Our results show that the expression of various growth factors was increased in the HA-CA+ADSCs group compared with ADSCs or HA-CA only treatment groups. If a large number of ADSCs were not present (survived) in the HA-CA+ADSCs group, the expression of growth factors should be similar to those of the ADSCs or HA-CA group. Therefore, we think that more hADSCs exist (or survive) in the HA-CA+ADSCs group.

- Mice were sacrificed on day 14 for PKH26 labeled ADSC tracking. Briefly, hADSCs were harvested and resuspended in 1 mL of diluent C solution. Then, 4 µL PKH26 dye was added followed by incubation for 5 min. FBS (1 mL) was added to quench for 2 min, followed by washing with PBS. We have now added this information in the Manuscript.

Please define exactly what tissue and how many replicates are analyzed by RT-PCR. A non-STZ treated control would also have been relevant.

- The number of tissue was 6 in each group (Figure 4 Legend), and all reactions were performed in triplicate (Material and Method).

- The Association for Assessment and Accreditation of Laboratory Animal Care International system recommends extremely strict and minimal animal sacrifices. We had a limit on the number of animals, so non-STZ treated control was not considered.

Please provide adequate quantification (and specify methodology) for Figure 4I to support that “VEGF expression was significantly increased” and for TUNEL assay in Figure 2C (i.e., Quantitative analysis of multiple sections throughout the wounded area using low magnification or stitched images).

- VEGF-positive cells were manually counted in five microscopic fields of each stained sample, and the mean value was statistically analyzed (one-way analysis of variance (ANOVA) followed by Dunn’s multiple comparison post-hoc tests). Bar graphs showing that mice in group G4 had the highest number of VEGF-positive cells (**< 0.01, n = 8 per group). Quantitative data were expressed as mean ± standard deviation; G1: 14.75 ± 4.41, G2: 24.75 ± 6.01, G3: 27.88 ± 4.93, G4: 46.3 8± 4.76. We have now mentioned this content in the Manuscript.

- The level of apoptosis was not significantly different among the groups. Quantitative data show the value of mean and standard deviation (G1: 18.71 ± 2.06, G2: 24.57 ± 4.15, G3: 21.00 ± 4.18, G4: 20.56 ± 3.34). We have now included this information in the Manuscript.

- The image resolution has been changed from 300dpi to 600dpi. (The journal system does not attach each Figure file but uploads it by inserting it into the MS word file.)

Please specify what n=8-10 means in Figure legends 2 and 3 (how many sections per wound/animal how were they spaced within the tissue, how many animals?).

- Forty C57BL/6 mice were equally divided into four groups. But, some mice were excluded from the study because of weight loss and blood glucose levels< 250 mg/dL.

When the authors report magnifications at x400 or x100, do they include only the objectives or is this using the oculars?

- We added the scale bar in all Figures.

Please specify “Compared with the control (DBW) group, skeletal muscle regeneration was higher. How did you quantify the Masson’s trichrome staining? The authors should document a non-wounded area as control adjacent to the wound in their current sections or in another animal.

- The opinion of skeletal muscle regeneration in Masson's trichrome staining is the result of observation of the stained sample. We observed all the stained samples and confirmed the regeneration of skeletal muscle in other groups than the control group. We tried quantitative analysis based on the reviewer's comments, however, we were not able to conduct color ratio analysis. Although we could not provide quantitative analysis results, the authors objectively observed stained samples and presented histopathological results instead. We will consider reviewers' comments such as non-STZ treatment group and non-wounded area in further study.

Round 2

Reviewer 1 Report

The authors made efforts to answer the reviewer’s comments in the limited time that they had.

They have specified multiple sections in the methodology that were lacking from the previous version of the manuscript. There is still a significant gap between the scope of the results that were obtained and the over-interpretation in their conclusions.

Major concerns are:

1/The authors comment on the role of inflammation in diabetic wounds in the introduction. The experimental model is using streptozotocin to induce diabetes by damaging pancreatic cells and promote hyperglycemia, but the wound is mechanical. While the authors provided new references for changes in the duration of wound healing, they need to either substantiate with references that skin wound regeneration in STZ-treated BL6 mice is modeling diabetic wounds or provide some evidence from their experiments, especially since the control group can achieve wound closure in 21 days and that there is no difference in local apoptosis. The reviewer previously suggested to detect immune cells (in the control mice). How does lack of dermis regeneration in the control group relate to the histopathology of the diabetic wound in patients?

2/ What is the aim of PKH26 labeling of ADSC? The authors need to either provide quantifiable data (migration of ADSC from injection site,…) or further characterize PKH26+ cells in the tissue. As presented, the data is only a technical validation of ADSC injection without even demonstrating survival.

3/ Related to comment #1. The authors claim that ADSC+HA-CA promoted angiogenesis, but in diabetic patients, endothelial cells are dysfunctional and angiogenesis is compromised. For instance, see Uzoagu et al “Compromised angiogenesis and vascular Integrity in impaired diabetic wound healing”, PLoS One 2020 for a mouse model of diabetic wound healing. The authors need to demonstrate similar dysfunctional endothelial cells in STZ-treated BL6 mice to validate the diabetic wound model and subsequently their regenerative model. The authors have to include a non-STZ-treated control to validate such effects on wound healing (for instance in Figure 3).

4/ The authors need to provide more extensive data for local pro-angiogenic expression and VEGF expression should be further characterized with mouse endothelial and human ADSC (PKH26, human antigens) markers. The RT-PCR panel is too limited to substantiate the schematic diagram in Figure 5. Additional validation with Western blot is highly recommended.

Reviewer 3 Report

The authors made efforts to answer the reviewer’s comments in the limited time that they had.

They have specified multiple sections in the methodology that were lacking from the previous version of the manuscript. There is still a significant gap between the scope of the results that were obtained and the over-interpretation in their conclusions.

Major concerns are:

1/The authors comment on the role of inflammation in diabetic wounds in the introduction. The experimental model is using streptozotocin to induce diabetes by damaging pancreatic cells and induce hyperglycemia, but the wound is mechanical. While the authors provided references for the duration of wound healing, they need to either substantiate with references that skin wound regeneration in STZ-treated BL6 mice is modeling diabetic wounds or provide some evidence from their experiments, especially since the control group can achieve wound closure in 21 days and that there is no difference in local apoptosis. The reviewer previously suggested to detect immune cells (in the control mice). How does lack of dermis regeneration in the control group relate to the histopathology of the diabetic wound in patients?

2/ What is the aim of PKH26 labeling of ADSC? The authors need to either provide quantification data (migration of ADSC from injection site,…) or further characterize PKH26+ cells in the tissue. As presented, the data is only a technical validation of ADSC injection without demonstrating survival.

3/ Related to comment #1. The authors claim that ADSC+HA-CA promoted angiogenesis, but in diabetic patients, endothelial cells are dysfunctional and angiogenesis is compromised. For instance, see Uzoagu et al “Compromised angiogenesis and vascular Integrity in impaired diabetic wound healing”, PLoS One 2020. The authors need to demonstrate similar dysfunctional endothelial cells in STZ-treated BL6 mice to validate the diabetic wound model and their regenerative model. The authors have to include a non-STZ-treated control to validate such effects on wound healing (for instance in Figure 3).

4/ The authors need to provide more extensive data for local pro-angiogenic expression and VEGF expression should be further characterized with mouse endothelial and human ADSC (PKH26, human antigens) markers. The RT-PCR panel is too limited to substantiate the schematic diagram in Figure 5. Additional validation with Western blot is highly recommended.